# Rate of force development in the quadriceps of individuals with severe knee osteoarthritis: A preliminary cross-sectional study

**Yusuke Suzuki**[1]*, **Hirotaka Iijima**[2,3], **Masatoshi Nakamura**[1], **Tomoki Aoyama**[2]

**1** Department of Physical Therapy, Faculty of Rehabilitation, Niigata University of Health and Welfare, Niigata, Japan, **2** Department of Physical Therapy, Human Health Sciences, Graduate School of Medicine, Kyoto University, Kyoto, Japan, **3** Department of System Design Engineering, Faculty of Science and Technology, Keio University, Yokohama, Japan

* yusuke-suzuki@nuhw.ac.jp

**Data Availability Statement:** All relevant data are within the paper and its Supporting information files.

## Abstract

Knee osteoarthritis (KOA) is a leading cause of knee pain and disability due to irreversible cartilage degeneration. Previous studies have not identified modifiable risk factors for KOA. In this preliminary cross-sectional study, we aimed to test the following hypotheses: individuals with severe KOA would have a significantly lower quadriceps rate of force development (RFD) than individuals with early KOA, and the decrease in quadriceps RFD would be greater than the decrease in maximum quadriceps strength in individuals with severe KOA. The maximum isometric strength of the quadriceps was assessed in individuals with mild (Kellgren and Lawrence [K&L] grade 1–2) and severe KOA (K&L grade 3–4) using a hand-held dynamometer. The RFD was analyzed at 200 ms from torque onset and normalized to the body mass and maximum voluntary isometric contraction torque. To test whether the quadriceps RFD was lowered and whether the lower in the quadriceps RFD was greater than the lower in maximum quadriceps strength in individuals with severe knee OA, the Mann–Whitney U-test and analysis of covariance were performed, respectively. The effect size (ES) based on Hedges' g with a 95% confidence interval (CI) was calculated for the quadriceps RFD and maximum quadriceps strength. Sixty-six participants were analyzed. Individuals with severe KOA displayed significantly lower quadriceps RFD (p = 0.009), the lower being greater than the lower in maximum quadriceps strength (between-group difference, ES: 0.88, -1.07 vs. 0.06, -0.22). Our results suggest that a decreased quadriceps RFD is a modifiable risk factor for progressive KOA. Our finding could help in the early detection and prevention of severe KOA.

## Introduction

Knee osteoarthritis (KOA) is a common form of arthritis and a leading cause of knee pain and disability due to irreversible cartilage degeneration [1]. Deterioration of physical function is common among individuals with KOA [2]. However, no radical treatment is currently

**Funding:** This work was supported by JSPS KAKENHI with grant number 20K19356 (Yusuke Suzuki). The funders had no role in study design, data collection and analysis, decision to publish, or preparation of the manuscript.

**Competing interests:** The authors have declared that no competing interests exist.

available for KOA, and no treatment approaches for controlling KOA progression have been reported. Although risk factors of KOA were identified in a previous review [3], the prognostic factors identified in this review (having KOA and a high serum level of hyaluronic acid) are not modifiable and can only be used to identify patients at high risk of disease progression. Identifying and updating modifiable risk factors that can be improved by rehabilitation, such as muscle strength is, therefore, a critical unmet need.

Skeletal muscles provide shock absorption and distribute the load across the joint [4]. Failure of the protective mechanisms from weakened quadriceps muscle strength can lead to harmful load distribution in the knee joint [5]. Therefore, the quadriceps muscle functions as a general shock absorber to protect articular knee joint surfaces during loading. However, reports on the relationship between maximum quadriceps strength and radiographic severity, which reflects reduced cartilage thickness of the knee joint in patients with KOA, have been equivocal. A systematic review and meta-analysis showed the quadriceps muscle weakness at baseline to be a risk factor for later radiographic KOA [6]. In contrast, a meta-analysis reported that knee extension strength is not associated with the radiographic severity of KOA [7]. A prospective cohort study reported that greater quadriceps strength did not influence cartilage loss at the tibiofemoral joint in KOA [8]. Therefore, it may be insufficient to solely focus on maximum quadriceps strength to prevent the progression of KOA.

The rate of force development (RFD) is an index reflecting explosive muscle strength. The RFD has recently received attention as a measure of muscle function, distinct from maximum strength. In terms of muscle contraction time, maximum strength requires more than 300 ms for exertion. Muscle activity in daily life is performed over a muscle contraction time until 50–200 ms [9, 10]. Therefore, with regard to muscle contraction time, the RFD may be more closely related to daily activities than maximum strength. For example, previous study reported that the RFD was related to elbow movement performance [11]. Moreover, studies have reported that the RFD in KOA patients was significantly associated with the activities of daily living score [12] and biomechanical gait variables [13]. These findings collectively suggest that the RFD is more closely associated with daily activities than maximal strength. Moreover, patients with radiographically severe KOA (Kellgren and Lawrence [K&L] grade > 2) have significantly more impairments in daily activities than mild KOA patients (K&L grade 1–2) [14]. Thus, it is plausible that patients with radiographically severe KOA may have a more significantly diminished quadriceps RFD than maximum quadriceps strength. However, the relationship between RFD and radiographic KOA severity has not yet been investigated.

The aim of the preliminary study is to test the hypothesis that: (1) individuals with severe KOA would have a significantly lower quadriceps RFD than individuals with early KOA and (2) the diminish in quadriceps RFD would be greater than the diminish in maximum quadriceps strength in individuals with severe KOA. If the trend of the relationship between RFD and radiographic KOA can be clarified in a preliminary study with a small sample, it will be important information for early detection of severe KOA development and prevention of progression to severe KOA.

## Materials and methods

### Participants

This preliminary study employed a cross-sectional design. Elderly participants who reported current knee pain were identified via a mailed survey and were invited to visit a research center in Kyoto in September 2019. This study was conducted in accordance with the Declaration of Helsinki and was approved by the ethics committee of Kyoto University (approval no. R2151-1). Written informed consent was obtained from all participants before their enrolment. All

participants had a history of pain in one or both knees. The eligibility criteria were (1) age ≥ 45 years, (2) knees with early and severe osteoarthritis (OA) (i.e., K&L grade 1–2 and 3–4, respectively, according to the original version [15]) in one or both knees in the medial tibiofemoral compartment as evaluated using weight-bearing anteroposterior radiographs, and (3) ability to walk independently on a flat surface without the use of any ambulatory assistive device. Participants with bilateral KOA were not considered separately from those with unilateral KOA because it was necessary to increase the sample size. The exclusion criteria were (1) a history of knee surgery, (2) rheumatoid arthritis, (3) periarticular fracture, (4) concurrent neurological problems, or (5) knees with pre-radiographic OA (i.e., K&L grade 0).

## Measurements

For all participants, demographic data and knee pain were evaluated as individual characteristics and covariates. Outcome measures were radiographic evaluation and measurement of maximum quadriceps strength and the RFD.

## Demographic data and knee pain

Data on age, sex, and height were self-reported by participants. Weight was measured on a scale, with the participants clothed but barefoot. Body mass index was calculated by dividing weight in kilograms by the square of height in meters. Knee pain over previous several days was evaluated using a VAS.

## Radiographic evaluation

Anteroposterior radiographs of both knees in the fully extended weight-bearing and foot map positions were obtained at enrolment. The radiographic severity of the medial compartment in the tibiofemoral joint was assessed by a trained examiner. The K&L grade was scored as follows: 0, normal; 1, doubtful joint space narrowing (JSN) and possible osteophyte; 2, definite osteophyte and possible JSN; 3, multiple osteophytes, definite JSN, some sclerosis, and possible deformity of bone ends; and 4, large osteophyte, marked JSN, severe sclerosis, and definite deformity of bone ends. The intra- and inter-rater agreements for the K&L grade determination were excellent (intra-rater: $\kappa = 0.88$, 95% CI = 0.83–0.92; inter-rater: $\kappa = 0.84$, 95% CI = 0.79–0.90) [16]. To permit analysis the association of outcome measures with radiographic severity, the present sample was classified, as reported previously [17], into mild KOA (K&L grade 1–2) and severe KOA (K&L grade 3–4).

## Maximum quadriceps strength and RFD

The maximum voluntary isometric contraction (MVC) in both legs was measured using a handheld dynamometer (HHD) (Mobie, Sakai Medical Co., Ltd, Japan; accuracy: ± 2% rated output), in accordance with a previously validated method for community-dwelling elderly patients prone to falling [18, 19]. The HHD is a simple tool for objectively quantifying muscle strength and is widely used in clinical practice. Participants were instructed to remain seated in an upright position, and the knee was placed at 90˚ flexion. The HHD was attached 10 cm proximal to the lateral malleolus and held in place with an inelastic strap looped around the therapy bed and fastened. The length of the strap allows for isometric contraction with the knee at 90˚ flexion during testing. Participants were instructed to extend their legs for 5 seconds as fast and as hard as possible. A few practice attempts were made before the actual measurement. Strong verbal encouragement was provided to ensure maximal effort. There were no complaints of knee pain from the participants during the RFD and maximal muscle

strength measurements. The results of quadriceps strength test were recorded in Newtons (N), and two repetitions were performed. Force measures were acquired and passed through an analog-to-digital converter (Power Lab, AD Instruments, Australia), sampled at 1,000 Hz, and plotted using LabChart 8 software (AD Instruments, Australia). The distances from the center of the force pad to the knee were recorded for each participant and used to convert measured forces into joint torques. Torque values were calculated by multiplying the obtained force values by the distances from the center of the force pad to the knee. MVC in the quadriceps was defined as the maximal torque value obtained from two attempts. The MVC was normalized to body mass. We assessed the RFD over the first 200 ms of maximal isometric contraction, whereby the onset of contraction was deemed as the point at which torque had increased 4 Newton-meter (Nm) above the baseline value [20]. The RFD was analyzed at 200 ms from the onset of torque [21]. The RFD was normalized to body mass and MVC torque (%MVC) [20, 22]. The average of two repetitions was used for statistical analysis.

## Statistical analysis

To minimize any bias introduced by similarities between the right and left knees of the same participants, only one knee per participant (index knee) was analyzed. The index knee was defined as the more painful knee currently or in the past. For participants who reported equally painful knees, the index knee was selected randomly using a computer-generated permuted block randomization scheme [23].

The characteristics of the participants are presented as mean ± standard deviation (SD) for continuous variables and as numbers and percentages for nominal and ordinal variables. The Mann–Whitney U-test was used to determine group differences between participants with mild and severe KOA because the dependent variable was not normally distributed as determined by the Shapiro-Wilk test.

To test the hypothesis that individuals with severe KOA would display a significantly decreased quadriceps RFD, the Mann–Whitney U-test was used to assess the differences between participants with mild and severe KOA.

Moreover, to test that the results of the Mann–Whitney U-test did not change after adjusting for covariates and to test the hypothesis that the decreased quadriceps RFD would be greater than the decrease in maximum quadriceps strength in individuals with severe KOA, the following posthoc analysis was performed. To compare between-group differences for mild and severe KOA, an independent measures analysis of covariance was performed, with age (years), sex, and knee pain VAS (mm) as covariates. The effect size (ES), Hedges' g, was calculated as the standardized mean difference (SMD) with mean ± SD and 95% CI using Review Manager Software (RevMan 5.3; Cochrane Collaboration, Oxford, UK) for the quadriceps RFD and maximum quadriceps strength. The Cohen scale was used to interpret ES, where 0.2 represents a small effect, 0.5 a moderate effect, and 0.8 a large effect. Data analyses were performed using JMP Pro 15.0 (SAS Institute, Cary, NC, USA). A p-value of < 0.05 was considered significant.

## Results

In total, 150 participants were recruited. Of these, 69 (46.0%) were excluded due to the absence of pain in the knee; four (2.6%) due to pre-radiographic KOA; and 11 (7.3%) due to missing data. A total of 66 participants were included in the final analysis (Fig 1). Table 1 shows the group differences between participants with mild and severe KOA. Fig 2 shows the typical isometric force-time curve indicating the maximum strength and RFD in mild and severe KOA.

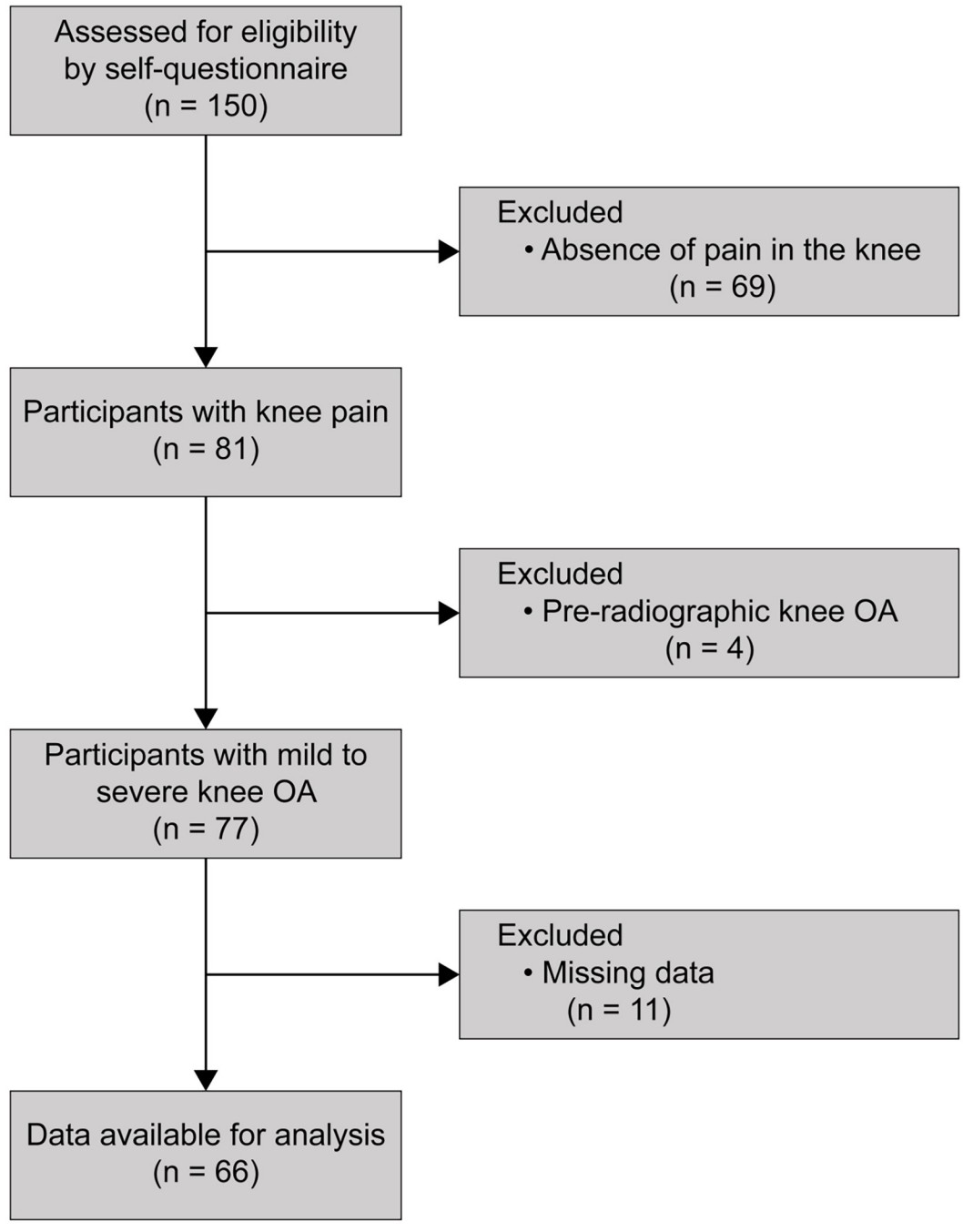

**Fig 1. Flow diagram of the study protocol.**

### Decreased quadriceps RFD in severe KOA

Participants with severe KOA had a significantly lower quadriceps RFD (p = 0.009) but not maximum strength (p = 0.50) when compared to group with mild KOA (Table 2).

When adjusted for age, sex, and knee pain VAS, participants with severe KOA exhibited a significantly decreased quadriceps RFD (between-group difference: 0.88; 95% CI: 0.23 to 1.54). The quadriceps RFD exhibited a large effect (ES: -1.07; 95% CI: -1.83 to -0.30) (S1 Table).

**Table 1. Characteristics of participants with mild and severe KOA.**

| | Mild KOA* | Severe KOA* | p-value† |
|---|---|---|---|
| | (n = 58) | (n = 8) | |
| Age (years) | 74.05 ± 5.09 | 75.87 ± 4.45 | 0.23 |
| Women, n (%) | 50 (86.2) | 6 (75.0) | 0.59 |
| Height (cm) | 154.41±8.73 | 157.75±10.84 | 0.23 |
| Weight (kg) | 53.43±9.08 | 56.37±7.02 | 0.24 |
| BMI (kg/m$^2$) | 22.35 ± 2.91 | 22.75 ± 2.72 | 0.62 |
| K&L grade, n (%) | | | |
| Grade 1 | 17 (29.3) | – | |
| Grade 2 | 41 (70.7) | – | |
| Grade 3 | – | 6 (75.0) | |
| Grade 4 | – | 2 (25.0) | |
| Knee pain VAS (mm) | 25.06 ± 22.05 | 29.0 ± 23.13 | 0.56 |

BMI: body mass index; KOA: knee osteoarthritis; K&L grade: Kellgren and Lawrence grade; VAS: visual analog scale

* Values are expressed as mean ± SD or number (percentage)

† Based on the unadjusted analysis (Mann–Whitney U-test) of participants with early and severe KOA

When adjusted for age, sex, and knee pain VAS, participants with severe KOA did not exhibit significantly decreased maximum quadriceps strength (between-group difference: 0.06; 95% CI: -0.09 to 0.21). Maximum quadriceps strength exerted a small effect (ES: -0.22; 95% CI: -0.96 to 0.52) (S2 Table).

## Discussion

Findings of this preliminary study supported our hypotheses and demonstrated that individuals with severe KOA displayed a significantly lower quadriceps RFD than individuals with early KOA. Moreover, the difference in the quadriceps RFD was greater than the difference in maximum quadriceps strength in individuals with severe KOA based on a comparison of the ES between the quadriceps RFD and maximum quadriceps strength.

The mechanism of the early (range 0–100 ms) RFD was associated with neural drive, contractile properties, and fiber type composition, and the late (range 0–200 ms) RFD was associated with muscle size, muscle strength, neural drive, and stiffness of the tendon-aponeurosis complex [24]. Regarding the fiber type, skeletal muscle fibers are broadly classified as "slow-twitch" (type I) and "fast-twitch" (type II). The RFD is an index that reflects explosive muscle strength and is thus indicative of type II muscle fibers [25]. Type II muscle fibers are more affected by age-related atrophy than type I muscle fibers [26]. However, atrophy of type II muscle fiber has not been observed in patients with severe KOA compared with that in age-matched elderly controls [25]. Moreover, we did not observe a significant reduction in early (range 0–100 ms) RFD in severe KOA patients in this study (S3 Table). Therefore, the involvement of fiber type in this study is considered small. Regarding the neural drive, neural activation and rapid neuromuscular activation of the quadriceps were impaired in KOA patients [27, 28]. Moreover, in this study, RFD was adjusted for peak torque; therefore, the effects of muscle size and muscle strength were considered to have been excluded. For this reason, individuals with severe KOA displayed a significantly decreased quadriceps RFD, and the quadriceps RFD exhibited a large ES due to the effect of neural drive and the stiffness of the tendon-aponeurosis complex.

In this study, participants with severe KOA did not exhibit significantly differences in the maximum quadriceps strength. A systematic review reported that decreased maximum

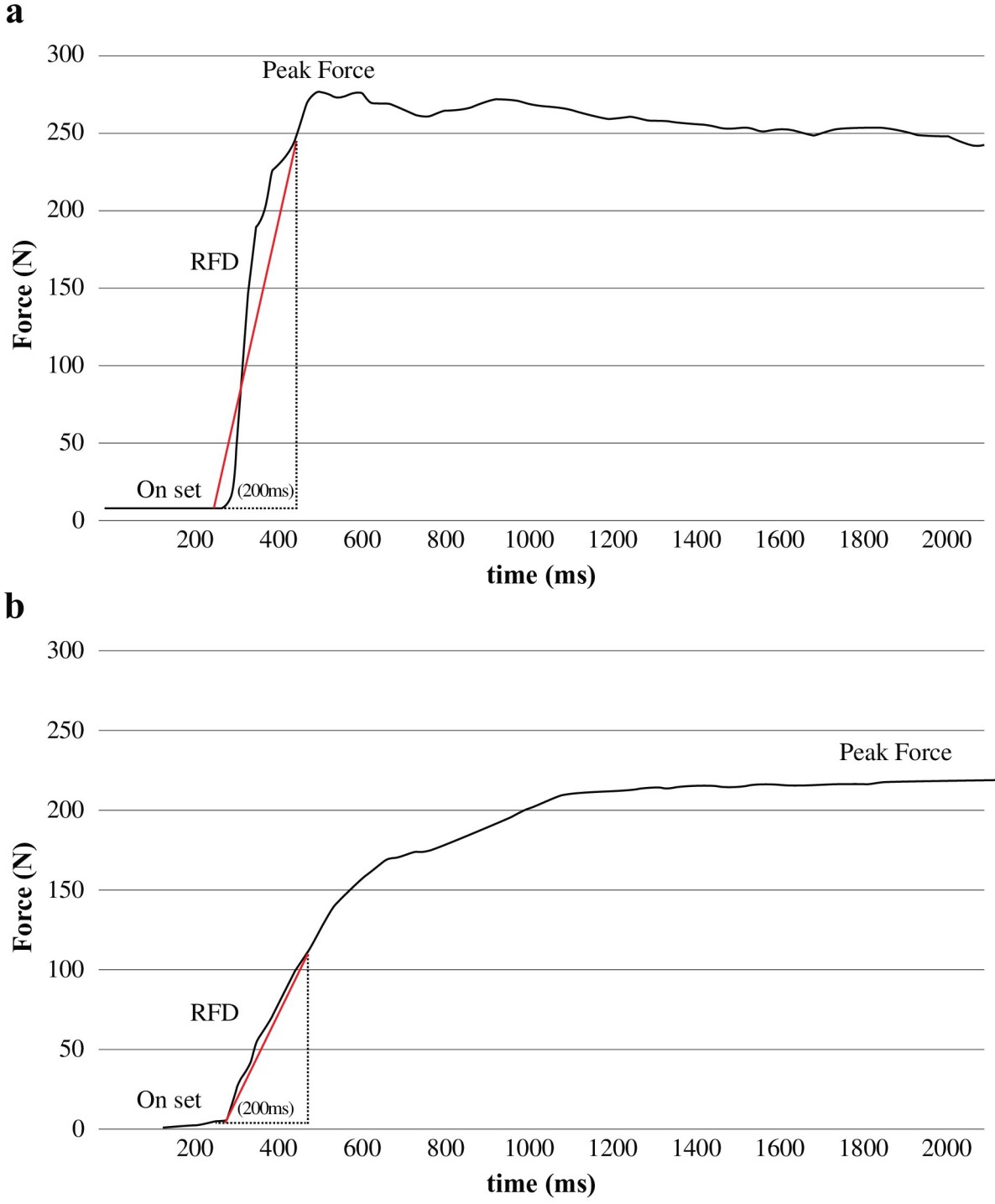

**Fig 2.** (a) Isometric force-time curve indicating the maximum strength and RFD in individuals with mild KOA. (b) Isometric force-time curve indicating the maximum strength and RFD in individuals with severe KOA. KOA; knee osteoarthritis; RFD; rate of force development.

quadriceps strength was associated with an increased risk of symptomatic and functional deterioration but not radiographic tibiofemoral JSN [7]. The results of the maximum quadriceps strength in this study support this review.

**Table 2. Differences between participants with mild and severe KOA in the quadriceps RFD and maximum quadriceps strength.**

| | Mild KOA* | Severe KOA* | p-value† |
|---|---|---|---|
| | (n = 58) | (n = 8) | |
| Quadriceps RFD (%MVC/ms*kg) | 6.97±1.72 | 5.10±1.71 | **0.009** |
| Maximum quadriceps strength (Nm/kg) | 1.49±0.41 | 1.39±0.69 | 0.50 |

KOA: knee osteoarthritis; RFD: rate of force development

* Values are expressed as mean ± SD or number (percentage)

† Based on the unadjusted analysis (Mann–Whitney U-test) between participants with early and severe KOA

The clinical implications of our findings are as follows. First, weakness in the quadriceps RFD may be related to the early detection of severe KOA development. Therefore, the development of severe KOA, which can be determined only by radiography or magnetic resonance imaging, may be easily determined by measuring the quadriceps RFD if causal relationships are identified in future investigations. Second, interventions for RFD of the quadriceps muscle may lead to the prevention of severe KOA. Since the weakened quadriceps muscle strength can lead to a failure of harmful load distribution in the knee joint [4, 5], the quadriceps muscle seem to be a general shock absorber during knee joint load. Moreover, excessive mechanical stress on knee cartilage due to muscle weakness has been suggested to contribute to degenerative processes [29]. In this study, quadriceps RFD was found to be more strongly related to severe KOA than maximal quadriceps strength. Therefore, if a causal relationship is found between quadriceps RFD and radiographic severity, improving the quadriceps RFD may lead to the prevention of severe KOA.

## Study limitations

This preliminary study has several limitations. First, the participants were motivated individuals given that they actively enrolled by responding to e-mails and were recruited via public relations magazine advertisements; thus, there may have been a selection bias. Second, since a power calculation was not performed, the relationship between severe KOA and weakness of maximum quadriceps strength cannot be determined. Third, since this study was preliminary, it included only a small number of individuals with severe KOA, which may bias the incidence of severe KOA events. Therefore, in this study, the Mann–Whitney U-test was performed, followed by analysis of covariance as a post hoc analysis. Moreover, women constitute a majority of the sample in this studied. Therefore, we conducted a sensitivity analysis to see if the results remained the same after excluding the males (S4 Table). However, it is necessary to increase the sample size and confirm the results with different analysis methods in the future. Finally, given the cross-sectional nature of this study, causal associations of severe KOA with the quadriceps RFD could not be determined. Further investigations, including prospective studies that clarify causal associations, are required to confirm our results.

## Conclusion

We demonstrated that individuals with severe KOA had significantly lower quadriceps RFD than individuals with early KOA. Moreover, the decrease in the quadriceps RFD was greater than the decrease in maximum quadriceps strength in individuals with severe KOA. These findings indicate that a decreased quadriceps RFD may be a modifiable risk factor for progressive KOA, which should be verified in future longitudinal studies.

## Supporting information

**S1 Table. Post hoc analysis to test the between-group differences adjusted for covariates in the quadriceps RFD for mild and severe KOA.**
(DOCX)

**S2 Table. Post hoc analysis to test the between-group differences adjusted for covariates in maximum quadriceps strength for mild and severe KOA.**
(DOCX)

**S3 Table. Differences between participants with mild and severe KOA in the quadriceps early (range 0–100 ms) RFD.**
(DOCX)

**S4 Table. Differences between women participants with mild and severe KOA in the quadriceps RFD and maximum quadriceps strength.**
(DOCX)

**S1 Checklist. STROBE statement—Checklist of items that should be included in reports of *cross-sectional studies*.**
(DOCX)

**S1 Data.**
(XLSX)

## Acknowledgments

The authors would like to thank the students of Human Health Sciences at Kyoto University for their help with data collection. We would like to thank Editage (www.editage.jp) for English language editing.

## Author Contributions

**Conceptualization:** Yusuke Suzuki.

**Data curation:** Yusuke Suzuki.

**Formal analysis:** Yusuke Suzuki, Hirotaka Iijima, Masatoshi Nakamura.

**Funding acquisition:** Yusuke Suzuki.

**Investigation:** Yusuke Suzuki, Hirotaka Iijima, Masatoshi Nakamura, Tomoki Aoyama.

**Methodology:** Yusuke Suzuki, Hirotaka Iijima, Masatoshi Nakamura.

**Project administration:** Yusuke Suzuki, Tomoki Aoyama.

**Supervision:** Tomoki Aoyama.

**Visualization:** Yusuke Suzuki, Hirotaka Iijima, Masatoshi Nakamura.

**Writing – original draft:** Yusuke Suzuki.

**Writing – review & editing:** Yusuke Suzuki, Hirotaka Iijima, Masatoshi Nakamura, Tomoki Aoyama.

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
