## [Decision Letter · Decision Letter 0]

23 Aug 2021

PONE-D-21-21042

Rate of force development in the quadriceps of individuals with severe knee osteoarthritis: a cross-sectional study

PLOS ONE

Dear Dr. Suzuki,

Thank you for submitting your manuscript to PLOS ONE. After careful consideration, we feel that it has merit but does not fully meet PLOS ONE’s publication criteria as it currently stands. Therefore, we invite you to submit a revised version of the manuscript that addresses the points raised during the review process.

We look forward to receiving your revised manuscript.

Kind regards,

Dragan Mirkov, Ph.D.

Academic Editor

PLOS ONE

Reviewers' comments:

Reviewer's Responses to Questions

**Comments to the Author**

1. Is the manuscript technically sound, and do the data support the conclusions?

Reviewer #1: Partly

2. Has the statistical analysis been performed appropriately and rigorously? 

Reviewer #1: No

3. Have the authors made all data underlying the findings in their manuscript fully available?

Reviewer #1: Yes

4. Is the manuscript presented in an intelligible fashion and written in standard English?

Reviewer #1: Yes

5. Review Comments to the Author

Reviewer #1: RFD in different conditions (post-injury, post-surgery, KOA or hip OA related etc.) is always topic of high research and clinical interest. The authors should be commended for their work. The concerns I have are the same ones the authors mentioned in study limitations. One of major ones is the sample size, specifically, very small sample in the severe KOA group. I understand most of the time it is hard to increase the number of participants but since this is a cross-sectional study, I believe that is plausible. Nevertheless, the authors should be given a chance to provide substantial revisions of their work before being accepted for publishing.

Specific comments:

Title: consider adding word “preliminary” before cross-sectional study

Line 36. not decreased but lower, since this is a cross-sectional study

Line 52: please elaborate modifiable risk factors in more details

Line 68: it depends, some measures of RFD are largely independent from MVC torque thus represent another muscle function quality. For that reason, I here suggest using term „muscle function“rather than „muscle strength“

Line 70: do you imply on maximum RFD? To my modest knowledge, that is a bit slow since maximum RFD is usually exerted well below 200 ms. Individuals recovering from ACL reconstructions reach maximum RFD at approximately 180 ms from the contraction onset. You need to specify that times stated are observed for older adults and those with KOA.

Line 73: in facts, there are numerous papers published on that matter (check Mirkov et al. 2004)

Line 78: „in“ instead of „of“

Line 79: Consider beginning the sentence like “Thus, it is plausible that patients…”

Line 80: Consider using term „diminished“ than „decreased“ throughout the manuscript. It may be more suitable since this is cross-sectional study

Line 82: Please specify the aim(s) of the study.

Line 84: Please rewrite (see comment for Line 80)

Line 125: analysis instead of analyze

Lines 141-142: According to this sentence one may understand that RFD and MVC were assessed separately, when in fact you used a single test (MVIC performed as strong and as fast as possible) to assess both features of muscle function. Please rewrite.

Line 143: “Two repetitions were performed for each test” – again, it implies to something that wasn’t’ done

Line 151: Consider moving this sentence before the last one from previous paragraph.

Line 154: This sentence is redundant

Line 155: Since you presented muscle strength as torque (Nm) why simply not presenting RFD data as RTD, and then continue with eliminating influence of body mass and MVC.

Line 157: Why reporting reliability data when this was among your hypotheses? Reliability of HHD has been show by others as good to excellent.

Lines 177-179: it is not clear why the authors ran the ancova. It is generally recommended that categorical variables as gender and age not to be used as covariates. For example, women constitute a majority of the sample studied. Why not simply remove all the males. I don’t think it would change your findings since you have only 2 males in the severe KOA group, but it would be much easier for future readers to comprehend the results and the findings. On the other hand, both MVC strength and RFD data were normalized with respect to body mass, and shin length. One more reason not to consider gender as covariate.

Line 179: please put “effect size” before the abbreviation (ES)

Line 180: please provide interpretation for Hedge’s g

Table 1: for the sake of clarity add (%) in the “K & L grade” cell name

Line 210-212: lower seems more appropriate than decreased since you were testing significance of differences. Also, something like “Participants with severe KOA had significantly lower quadriceps RFD (p=0.009) but not maximum strength (p=0.50) when compared to group with mild KOA (Table 2).” would read easier.

Table 2: unit for RFD doesn’t seem correct, please double-check

Table 2 shows MVC data normalized to body mass but such detail is missing from the methods section

Lines 227-235: Will all the respect to authors effort and work, this section is redundant it they should consider removing it. It does not provide any new information than the previous paragraph but only confirms the results of Man-Whitney test

Lines 260-262: not decrease but difference. This study shows no difference between mild and severe KOA in maximum strength, but significant difference in RFD.

Line 281: Such statement would make sense if this was longitudinal study or if you had age/gender/physical activity etc. -matched controls. Maybe, just maybe, the maximum strength deteriorates to its lowest at stage of mild KOA and only RFD keeps deteriorating from there on (please refer to lines 270-271 where you discuss the influence of fiber type)

Lines 285-287 only repeat what was already said in lines 281-282

Lines 288-290: The positioning of such statement is not clear. It only repeats what was already stated above.

6. PLOS authors have the option to publish the peer review history of their article (what does this mean?). If published, this will include your full peer review and any attached files.

Reviewer #1: No

---

## [Author Response · Author response to Decision Letter 0]

6 Nov 2021

Dear Reviewer

Thank you for your insightful and helpful suggestions. We have made some corrections in accordance with the comments from reviewer, which hopefully meet the reviewer’s criteria. Revisions in the manuscript are shown in red font.

Sincerely yours,

Yusuke Suzuki

---

## [Decision Letter · Decision Letter 1]

27 Dec 2021

Rate of force development in the quadriceps of individuals with severe knee osteoarthritis: a preliminary cross-sectional study

PONE-D-21-21042R1

Dear Dr. Suzuki,

We’re pleased to inform you that your manuscript has been judged scientifically suitable for publication and will be formally accepted for publication once it meets all outstanding technical requirements.

Kind regards,

Emiliano Cè

Academic Editor

PLOS ONE

Additional Editor Comments (optional):

Reviewers' comments:

Reviewer's Responses to Questions

**Comments to the Author**

1. If the authors have adequately addressed your comments raised in a previous round of review and you feel that this manuscript is now acceptable for publication, you may indicate that here to bypass the “Comments to the Author” section, enter your conflict of interest statement in the “Confidential to Editor” section, and submit your "Accept" recommendation.

Reviewer #1: All comments have been addressed

2. Is the manuscript technically sound, and do the data support the conclusions?

Reviewer #1: Yes

3. Has the statistical analysis been performed appropriately and rigorously? 

Reviewer #1: Yes

4. Have the authors made all data underlying the findings in their manuscript fully available?

Reviewer #1: Yes

5. Is the manuscript presented in an intelligible fashion and written in standard English?

Reviewer #1: Yes

6. Review Comments to the Author

Reviewer #1: (No Response)

7. PLOS authors have the option to publish the peer review history of their article (what does this mean?). If published, this will include your full peer review and any attached files.

Reviewer #1: No

---

## [Editor Report · Acceptance letter]

3 Jan 2022

PONE-D-21-21042R1 

Rate of force development in the quadriceps of individuals with severe knee osteoarthritis: a preliminary cross-sectional study 

Dear Dr. Suzuki:

I'm pleased to inform you that your manuscript has been deemed suitable for publication in PLOS ONE. Congratulations! Your manuscript is now with our production department. 

Kind regards, 

on behalf of

Professor Emiliano Cè 

Academic Editor

PLOS ONE